# Image-level Memorization Detection via Inversion-based Inference Perturbation

**Yue Jiang**[1,2†]   **Haokun Lin**[1,2,3†]   **Yang Bai**[4]   **Bo Peng**[1]   **Zhili Liu**[5]
**Yueming Lyu**[6]   **Yong Yang**[4*]   **Xing Zheng**[4]   **Jing Dong**[1*]
[1] NLPR, MAIS, Institute of Automation, Chinese Academy of Sciences
[2] School of Artificial Intelligence, UCAS   [3] City University of Hong Kong
[4] Tencent Security Platform Department   [5] HKUST   [6] Nanjing University
```
{yue.jiang, haokun.lin}@cripac.ia.ac.cn
coolcyang@tencent.com  jdong@nlpr.ia.ac.cn
```

## Abstract

Recent studies have discovered that widely used text-to-image diffusion models can replicate training samples during image generation, a phenomenon known as memorization. Existing detection methods primarily focus on identifying memorized prompts. However, in real-world scenarios, image owners may need to verify whether their proprietary or personal images have been memorized by the model, even in the absence of paired prompts or related metadata. We refer to this challenge as image-level memorization detection, where current methods relying on original prompts fall short. In this work, we uncover two characteristics of memorized images after perturbing the inference procedure: *lower similarity of the original images and larger magnitudes of TCNP*. Building on these insights, we propose Inversion-based Inference Perturbation (IIP), a new framework for image-level memorization detection. Our approach uses unconditional DDIM inversion to derive latent codes that contain core semantic information of original images and optimizes random prompt embeddings to introduce effective perturbation. Memorized images exhibit distinct characteristics within the proposed pipeline, providing a robust basis for detection. To support this task, we construct a comprehensive setup for the image-level memorization detection, carefully curating datasets to simulate realistic memorization scenarios. Using this setup, we evaluate our IIP framework across three different memorization settings, demonstrating its state-of-the-art performance in identifying memorized images in various settings, even in the presence of data augmentation attacks. Our code and datasets are available at `https://github.com/joellejiang/IIP`.

## 1 Introduction

In recent years, text-to-image diffusion models such as Stable Diffusion (SD) (Rombach et al., 2022) and DALL-E (Ramesh et al., 2021) have achieved remarkable advancements, showcasing impressive generation fidelity and diversity when conditioned on given prompts. However, these models have also been found to reproduce partial or complete elements of their training images, a phenomenon broadly known as memorization (Schuhmann et al., 2022; Carlini et al., 2023; Somepalli et al., 2023a;b). Existing memorization detecting methods primarily focus on identifying whether specific prompts lead to memorized outputs (Wen et al., 2024; Ren et al., 2024) (Fig. 1 (Top)). The proposed magnitude of text-conditional noise prediction (MTCNP) (Wen et al., 2024) can effectively detect memorized prompts during the generation procedure (Fig. 2 (Top)).

However, in real-world scenarios, users may encounter situations where they need to verify whether a specific image has been memorized, either in partial or whole, by widely used text-to-image diffusion models. For instance, an image owner might wish to confirm whether their proprietary or personal content has been embedded into the model's memory, potentially violating copyright or

---

[†] Equal contribution.
[*] Corresponding authors.

privacy rights. Unlike existing approaches that rely on textual prompts that generate the memorized images, this scenario requires a direct, image-level assessment, as shown in Fig. 1 (bottom). Such a setting assumes that only the image itself is accessible, without information about the original paired prompts or associated metadata, making existing detection methods for memorized prompts (Wen et al., 2024; Ren et al., 2024) struggle to address this challenge. Ma et al. (2024) analyzes the detection of memorized images in diffusion models and identifies three conditions for memorization. Nevertheless, they focus solely on exact duplication with a relatively small subset, which limits its applicability in real-world scenarios.

In this paper, we reveal two characteristics of memorized images, which provide the basis for our image-level memorization detection. We discover that memorized images **exhibit lower similarity of the original images** and **maintain larger magnitudes of text-conditional noise predictions (MTCNP)** (Wen et al., 2024) after perturbation of the generation procedure.

Based on these findings, we propose a novel image-level memorization detection framework IIP free of original prompts. We are motivated to obtain a latent code with primary semantic information of original images and employ a perturbed inference procedure subsequently (as illustrated in Fig. 2(bottom)) This perturbation aims to derive both lower similarity of the original images and larger magnitudes of TCNP for memorized images, starkly contrasting with that of non-memorized images. Specifically, we employ unconditional DDIM Inversion (Song et al., 2020) at smaller timesteps to obtain semantic latent codes and perturb the inference procedure with optimized random prompts, which effectively derive significant characteristics for image-level memorization detection.

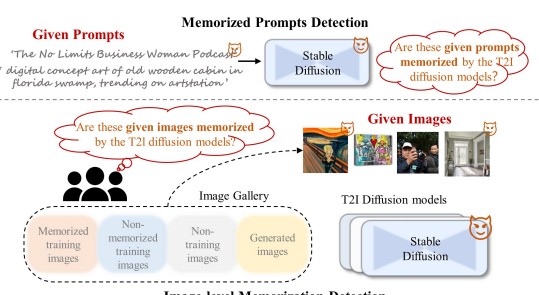

Figure 1: Differences between memorized prompts detection and image-level memorization detection.

We distinguish three different settings of memorization for comprehensive setup, including memorized vs non-memorized training images, memorized training vs non-training images (non-memorized), and the generated memorized vs non-memorized images. Furthermore, we simulate a variety of real-world memorization scenarios for practical applicability, including memorized images with varying degrees of replication. Extensive experiments and analysis demonstrate our method consistently achieves state-of-the-art performance across various datasets and different open-source text-to-image diffusion models.

We summarize our contribution as follows:

- We formalize image-level memorization detection task for text-to-image diffusion models across various datasets with a comprehensive setup, which assists in inspecting the security of training images against memorization risks.

- We uncover two characteristics of memorized images under inference perturbation with random prompts. Based on these findings, we propose a novel Inversion-based Inference Perturbation (IIP) framework to effectively detect memorized images without accessing original prompts.

- Extensive experiments demonstrate that our method achieves state-of-the-art performance in detecting memorized images in various settings. Additionally, our IIP framework shows strong robustness against data augmentation attacks.

## 2 MOTIVATION

### 2.1 PRELIMINARY

**Diffusion Model.** Diffusion models are probabilistic models (Ho et al., 2020) designed to learn the data distribution $p(x)$ by progressively denoising noises. In the text-conditioned latent diffusion

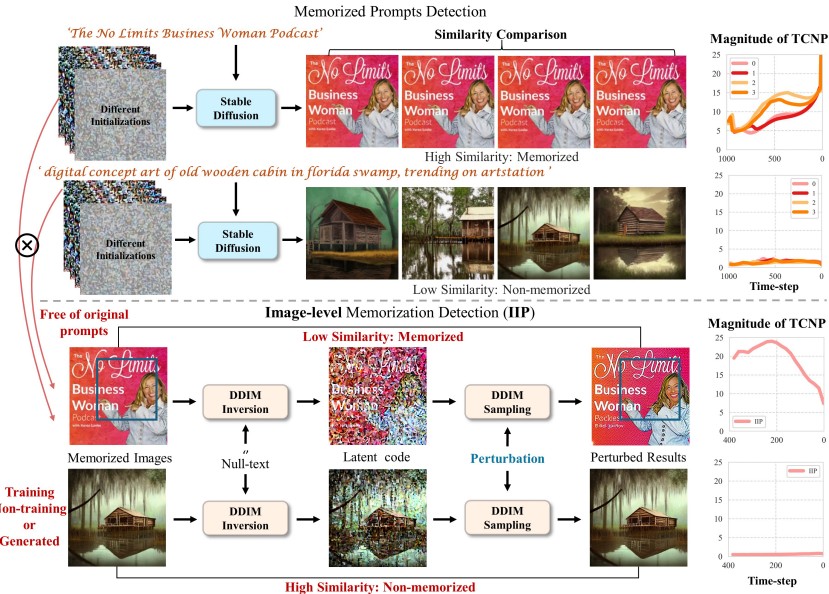

Figure 2: Memorized prompt detection methods (top) vs. our image-level memorization detection method IIP (bottom). Unlike previous methods, our approach focuses on identifying memorized images from a model without requiring access to the original prompts. Detection is achieved by evaluating the similarity of the original image and magnitude of TCNP values (Wen et al., 2024).

models, such as Stable Diffusion (Ramesh et al., 2022), the model learns the conditional latent distributions $p(z|c)$ using a denoising autoencoder $\epsilon_\theta(z_t, t, c)$, where $c$ denotes the conditioned text prompts. Based on image-text pairs, the conditional LDM Sohl-Dickstein et al. (2015); Ho et al. (2020) is trained as follows:

$$\mathcal{L}_{LDM} = \mathbb{E}_{E(x),c,t,\epsilon\sim\mathcal{N}(0,1)}[||\epsilon - \epsilon_\theta(z_t, t, c)||_2^2] \tag{1}$$

After training, the diffusion model generates images by employing classifier-free guidance (Ho & Salimans, 2022):

$$\widetilde{\epsilon}(z_t, t, c_p) = \epsilon_\theta(z_t, t, c_{null}) + s_g(\epsilon_\theta(z_t, t, c_p) - \epsilon_\theta(z_t, t, c_{null})), \tag{2}$$

Here, $z_t$ represents the latent code at timestep $t$, and $s_g$ denotes the guidance scale. The backbone $\epsilon_\theta$, derived from DDPM (Ronneberger et al., 2015; Ho et al., 2020), is used for noise prediction, where $c_p$ refers to the given prompt and $c_{null}$ represents the null-text prompt.

**Unconditional DDIM Inversion.** During the inference stage of diffusion models, the predicted noise undergoes further sampling to generate latent codes for previous timesteps. A commonly used method for this is DDIM sampling (Song et al., 2020), which follows a determined process for generating these latent codes. DDIM Inversion reverses the DDIM sampling process (from $t = 0$ to $T$) as follows:

$$z_{t+1} = \sqrt{\frac{\alpha_{t+1}}{\alpha_t}} \cdot (z_t - \sqrt{1 - \alpha_t} \cdot \epsilon_\theta(z_t, t, c)) + \sqrt{1 - \alpha_{t+1}} \cdot \epsilon_\theta(z_t, t, c) \tag{3}$$

Due to its deterministic nature, DDIM Inversion enables the retrieval of latent codes for a given image. Specifically, when using a null-text prompt $c_{null}$, referred to as unconditional DDIM Inversion, the inverted $z_T$ can reconstruct the original image with minimal error during the denoising process (Song et al., 2020; Mokady et al., 2023).

**Magnitude of text-conditional noise prediction.** Memorized prompts consistently guide the generation toward fixed results, regardless of the initialization, indicating significant text guidance during the denoising process. Wen et al. (2024) find that these prompts exhibit larger values of the latter term in Eq. 2 across a range of timesteps $T$, which are referred to as the magnitude of text-conditional

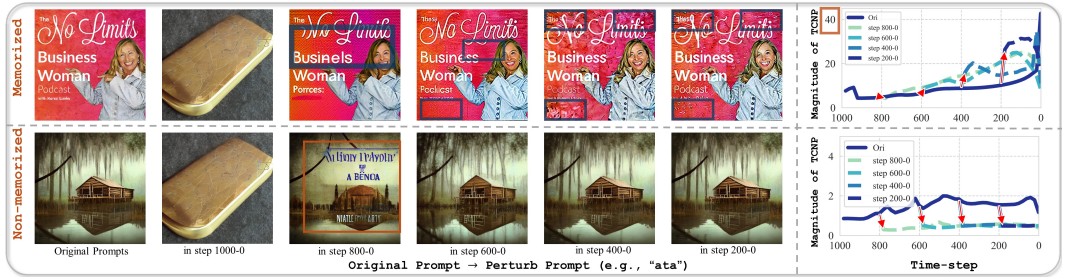

Figure 3: Perturbed results during different inference timesteps for memorized and non-memorized prompts. The magnitude of TCNP for perturbations is present on the right.

noise prediction (TCNP).

$$\frac{1}{T}\sum_{t=1}^{T}||\epsilon_\theta(z_t, t, c_p) - \epsilon_\theta(z_t, t, c_{null})||_2. \tag{4}$$

## 2.2 PERTURBATED ANALYSIS.

We conduct preliminary experiments to un-cover clues for identifying memorized images. Our approach primarily involves perturbed in-ference to explore potential features. Specifi-cally, we remove original prompts at different timesteps and replace them with meaningless ones (e.g., "ata") during the subsequent genera-tion period. Interestingly, we discover two key characteristics: after permutation operations, 1) the similarity of images changes in a distinct pattern, and 2) the MTCNP remains larger for memorized images.

**Memorized Images Exhibit Lower Similar-ity.** We apply perturbation during the genera-tion period, and the results of memorized and non-memorized images are shown in Fig. 3 (left). It can be observed that when the orig-inal prompts are replaced after *step 600*, the perturbed results display clear textual discrep-ancy of the original images. In contrast, non-memorized images show minimal discrepancy compared to their original images. Although in the latter inference stage, prompts reflect subtle information in most cases such as texture and color for diffusion models (Zhang et al., 2023b;

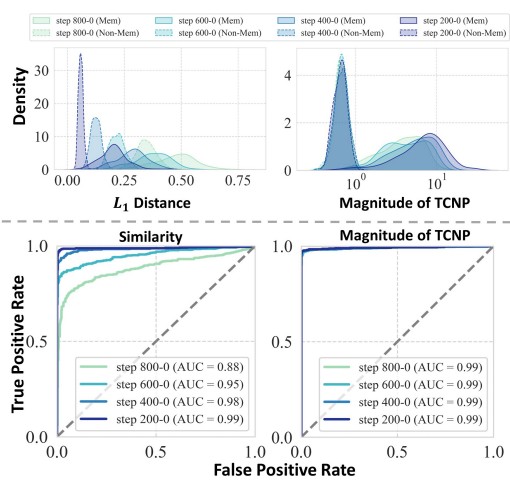

Figure 4: KDE (top) and ROC (bottom) curves of different metrics for perturbations across dif-ferent timesteps. Results are evaluated using 2000 images generated from memorized and non-memorized prompts following Wen et al. (2024).

Hertz et al., 2022), memorized image-text pairs would reveal greater discrepancy of the original results when prompts are exchanged during inference.

**Memorized Images Maintain Larger MTCNP.** We calculate the magnitude of TCNP before and after inference perturbation, with the results shown in Fig. 3 (right). As confirmed by Wen et al. (2024), original memorized prompts exhibit a significantly larger Magnitude of TCNP compared to non-memorized prompts. Our experiments extend this finding and reveal that even after applying meaningless prompt instead of the original prompt in the perturbed inference period, the magnitude of TCNP for memorized images remains consistently larger. This phenomenon suggests that a larger MTCNP does not solely depend on the original prompts, in the latter inference stage, the primary semantic information of an image is obtained, which also results in a larger MTCNP even without original prompts.

**Further Discussion.** To validate these characteristics, we further apply permutation on 2000 im-ages generated from 500 memorized and non-memorized prompts. The KDE plots of similarity

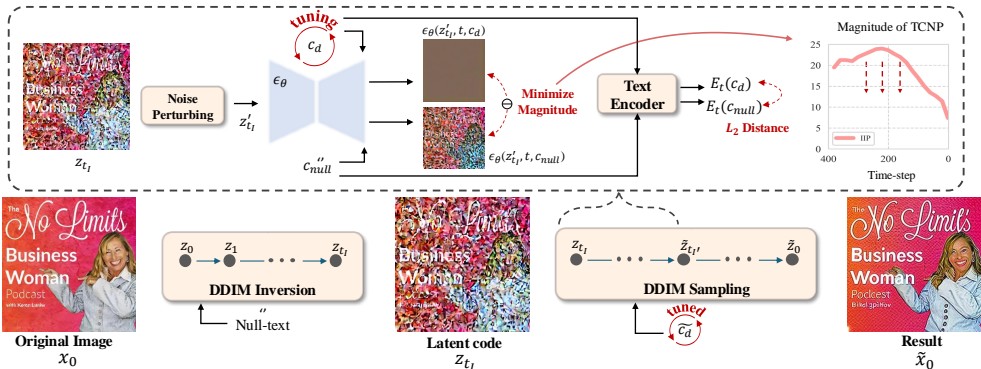

Figure 5: The overview of our IIP. We first employ unconditional DDIM Inversion to obtain a latent code with primary semantic information of the original image, and perturb the subsequent inference stage with optimized non-memorized prompt embedding. Our method allows memorized images to exhibit lower similarity of the original images, accompanied by larger magnitudes of TCNP.

degree and MTCNP are shown in Fig. 4 (top), with the $L_1$ score used as the similarity measure in the latent space. Our findings indicate that memorized results exhibit lower similarity compared to non-memorized results, while consistently maintaining larger MTCNP values. In addition, we apply these metrics to detect memorization in 2000 images, with the corresponding ROC plots presented in Fig. 4 (bottom). Both metrics effectively detect memorization, with AUC scores exceeding 0.9.

Based on these results, we are motivated to explore the use of latent codes that retain primary semantic information of the images in combination with random prompts, similar to the perturbed process starting from smaller timesteps. This procedure is likely to yield both lower of original images and larger MTCNP values for memorized images after the subsequent inference stage, which helps to effectively detect memorized images without original prompts.

## 3 METHOD

In this section, we provide a detailed introduction to our IIP. First, we employ unconditional DDIM Inversion to obtain latent codes containing primary semantic information from the original images. Next, we introduce optimized random prompt embeddings that are far from memorized during the subsequent inference stage. The perturbation is reinforced by minimizing the magnitude of TCNP to yield less memorized embeddings, thereby enhancing the significant characteristics for detection. The entire pipeline enables effective detection of memorized images free of original prompts, through evaluations of both the similarity between the reconstructed and original images in the latent space and the magnitude of TCNP during the inference stage.

### 3.1 DERIVE SEMANTIC LATENT CODE

We aim to obtain the latent code containing primary semantic and structural information of the original images, which leads to significant characteristics for memorization detection. DDIM Inversion (Song et al., 2020) is known for its ability to reverse the sampling process and derive latent codes of original images with varying prompts. In this work, we employ unconditional DDIM Inversion, which demonstrates the capability to nearly fully reconstruct the original images. Specifically, given the original image is $x_0$, the latent code $z_{t_I}$ at timestep $t_I$ is obtained as follows:

$$z_{t_I} = \sqrt{\frac{\alpha_{t_I}}{\alpha_{t_I-1}}} \cdot (z_{t_I-1} - \sqrt{1-\alpha_{t_I-1}} \cdot \epsilon_\theta(z_{t_I-1}, t, c_{null})) + \sqrt{1-\alpha_{t_I}} \cdot \epsilon_\theta(z_{t_I-1}, t, c_{null}), \quad (5)$$

where $c_{null}$ is the null-text prompt, $\epsilon_\theta$ is the noise predictor (Ho et al., 2020; Rombach et al., 2022). And $z_{t_I}$ is processed from $z_0 = E_i(x_0)$, and $E_i$ is the image encoder (Kingma, 2013) of the Diffusion Model (Rombach et al., 2022).

Moreover, to obtain more semantic information of the original images, we employ the inverted latent codes at a smaller timestep $t_I$, which shortens the inverted steps and also reduces inversion error.

## 3.2 OPTIMIZE NON-MEMORIZED PROMPT PERTURBATION

To derive significant characteristics for memorization detection, our goal is to perturb the subsequent inference started from the obtained latent code $z_{t_I}$ with prompts far from memorized. Given the random prompt is $c_d$ and subsequent inference stage starts from $z_{t_I}$, the magnitude of TCNP is calculated as follows:

$$||\epsilon_\theta(z_{t_I}, t, c_d) - \epsilon_\theta(z_{t_I}, t, c_{null})||_2. \tag{6}$$

To obtain the prompts far from memorized, we optimize the embedding of $c_d$ by minimizing the magnitude of TCNP across a range of timesteps (from $t_I$ to $t_{I'}$). To avoid straightforward optimization reductions in final magnitude and obtain an embedding of $c_d$ far from memorized, we add noises to obtain an approximate latent code $z'_{t_I}$ and constrain the embedding within a certain distance from the null-text embedding. The embedding of $c_d$ is optimized as follows:

$$L_{cd} = \frac{1}{t_I - t_{I'}} \lambda_d \cdot \sum_{t=t_{I'}}^{t_I} ||\epsilon_\theta(z'_t, t, c_d) - \epsilon_\theta(z'_t, t, c_{null})||_2 + \lambda_e \cdot ||E_t(c_d) - E_t(c_{null})||_2, \tag{7}$$

where $E_t$ is the text encoder (Kingma, 2013) of the Diffusion Model (Rombach et al., 2022), and $\lambda_d, \lambda_e$ are the hyper-parameters.

After optimization, we start inference using the latent code $z_{t_I}$ from timestep $t_I$ along with the optimized prompt embedding $\tilde{c}_d$. The subsequent latent codes are predicted following DDIM sampling (Song et al., 2020):

$$\tilde{z}_{t-1} = \sqrt{\frac{\alpha_{t-1}}{\alpha_t}} \cdot (\tilde{z}_t - \sqrt{1 - \alpha_t} \cdot \epsilon_\theta(\tilde{z}_t, t, \tilde{c}_d)) + \sqrt{1 - \alpha_{t-1}} \epsilon_\theta(\tilde{z}_t, t, \tilde{c}_d). \tag{8}$$

Finally, we can effectively detect memorized images without original prompts using both metrics of similarity to the original images and the magnitude of TCNP. Specifically, we find that $L_1$ distance in the latent space demonstrates greater effectiveness in our experiments, as it is sensitive to even subtle image discrepancies. The two metrics are calculated as follows:

$$\text{IIP}_{\text{sim}} = \sum_{i=1}^{N} |z_{0_i} - \tilde{z}_{0_i}|, \tag{9}$$

$$\text{IIP}_{\text{mtcnp}} = \frac{1}{t_I} \sum_{t=1}^{t_I} ||\epsilon_\theta(z_{t_I}, t, \tilde{c}_d) - \epsilon_\theta(z_t, t, c_{null})||_2, \tag{10}$$

where $z_0$ and $\tilde{z}_0$ are the latent codes of the original image and the generated image of our IIP.

## 4 EXPERIMENT

### 4.1 EXPERIMENTAL SETUP

**Datasets.** We evaluate our method across three dataset types for comprehensive analysis in various settings: 1) memorized and non-memorized training images, 2) memorized training images and non-training images, and 3) generated images from memorized and non-memorized prompts. All settings in the main paper are based on Stable Diffusion v1.4 (Rombach et al., 2022) and results on other versions of Stable Diffusion are provided in the appendix. For memorized training images, we follow Webster (2023) and collect 152 memorized images from the training set of SD v1.4. These images have an SSCD similarity score (Pizzi et al., 2022) exceeding 0.7, which strongly indicates duplication (Somepalli et al., 2023a;b). For non-memorized training images, we collect images from the member sets of LAION-mi and filter those with an SSCD similarity score lower than 0.15, retaining 1200 images. Since LAION-mi (Dubiński et al., 2024) extract part of the training set and non-member sets for Stable Diffusion v1.4, we select 1200 non-member images from LAION-mi as non-memorized non-training images. For generated images, we follow the setting of Wen et al. (2024), selecting 500 memorized prompts from Webster (2023) for Stable Diffusion v1.4 and 500 non-memorized prompts from various sources, including LAION (Schuhmann et al., 2022), COCO (Lin et al., 2014), Lexica.art, and randomly generated strings. Each prompt generates 4, 8, and 16

https://huggingface.co/datasets/Gustavosta/Stable-Diffusion-Prompts

images, resulting in a total of 2K, 4K, and 8K generated images for both types of prompts. We approximate that images generated from memorized prompts are highly memorized, albeit with slight differences. The following Table 1 presents the details of each category.

| Datset | Type | Number | Source | Filter |
|---|---|---|---|---|
| Memorized Training | Memorization | 152 | Training set of SDv1.4. | SSCD >0.7 |
| Non-memorized Training | Non-Memorization | 1200 | Member set of LAION-mi | SSCD <0.15 |
| Non-Training | Non-Memorization | 1200 | Non-member set of LAION-mi | / |
| Memorized Generated | Memorization | 2K / 4K / 8K | Memorized prompts | / |
| Non-memorized Generated | Non-Memorization | 2K / 4K / 8K | Non-memorized prompts | / |

Table 1: Detailed illustration of all datasets.

**Existing Baselines.** While no existing methods explicitly claim to detect memorization on image-level without paired prompts, we consider two types of baselines. First, methods that identify memorization by assessing the similarity between the original and reconstructed images using null-text prompts. For this, we select DDIM Inversion , image inpainting , and image-to-image generation as representative approaches. Specifically, for inpainting, we use a $16 \times 16$ grid mask that matches the original image size (Wu et al., 2024a). Second, membership inference attack methods serve as potential baselines, as they might detect memorization through stronger membership clues. We use two state-of-the-art methods, SecMI (Duan et al., 2023) and PIA (Kong et al., 2023), for comparison. All baselines, as well as our method, are implemented based on Stable Diffusion v1.4.

**Evaluation Metrics.** We calculate the Area Under the ROC Curve (AUC), and the True Positive Rate at the False Positive Rate of 1% (TPR@1%FPR) as metrics for all three datasets to evaluate the effectiveness of different methods. Additionally, because of the significant difference in the number of memorized and non-memorized images in both the training and non-training datasets, we compute the Area Under the PR Curve (AUC-PR) for these datasets. We compare the Accuracy (Acc.) for the generated datasets, which contain a balanced number of both types of images.

**Implementation Details.** All baselines employ the same DDIM sampling (Song et al., 2020), with inference steps and guidance scales consistently set to 50 and 7.5, respectively. Importantly, none of the methods access the original prompts for detection. In our experiment, we set $t_I = 20$ and $t_I' = 10$, and hyperparameters $\lambda_d = 1.0$, $\lambda_e = 1.0$. All experiments are conducted on a single A100.

## 4.2 EXPERIMENTAL RESULTS

| Method | Mem (Train) / Non-mem (Train) | | | Mem (Train) / Non-mem (Non-train) | | |
|---|---|---|---|---|---|---|
| | AUC | AUC-PR | TPR@1%FPR | AUC | AUC-PR | TPR@1%FPR |
| DDIM Inversion | 0.938 | 0.699 | 0.388 | 0.943 | 0.689 | 0.395 |
| Inpaint | 0.671 | 0.153 | 0.007 | 0.628 | 0.139 | 0.007 |
| Image-to-image | 0.721 | 0.170 | 0.000 | 0.798 | 0.227 | 0.000 |
| SecMI | 0.584 | 0.130 | 0.000 | 0.557 | 0.124 | 0.030 |
| PIA | 0.507 | 0.104 | 0.000 | 0.536 | 0.112 | 0.000 |
| **IIP$_{sim}$** | **0.999** | **0.995** | **0.987** | **1.000** | **0.998** | **0.993** |
| **IIP$_{mtcnp}$** | 0.995 | 0.962 | 0.842 | 0.997 | 0.976 | 0.875 |

Table 2: Quantitative results for different methods of detecting memorized images across different datasets. The best and second-best results are highlighted in **bold** and underline.

### 4.2.1 QUALITATIVE COMPARISON

Since SecMI and PIA primarily depend on the model's intermediate outputs and do not generate images, we conduct the qualitative comparison with reconstruction-based methods: DDIM Inversion, image inpainting, and image-to-image. Results are illustrated in Fig. 6. Notably, these baseline methods reconstruct images similar to original images for both memorized and non-memorized images, hardly determining memorization. In contrast, our method reveals significant textual inconsistencies in the reconstruction of memorized images, particularly in texture and certain colors,

```
https://huggingface.co/learn/diffusion-course/en/unit4/2
https://huggingface.co/docs/diffusers/en/api/pipelines/stable_
diffusion/img2img
https://huggingface.co/docs/diffusers/en/api/pipelines/auto_pipeline
```

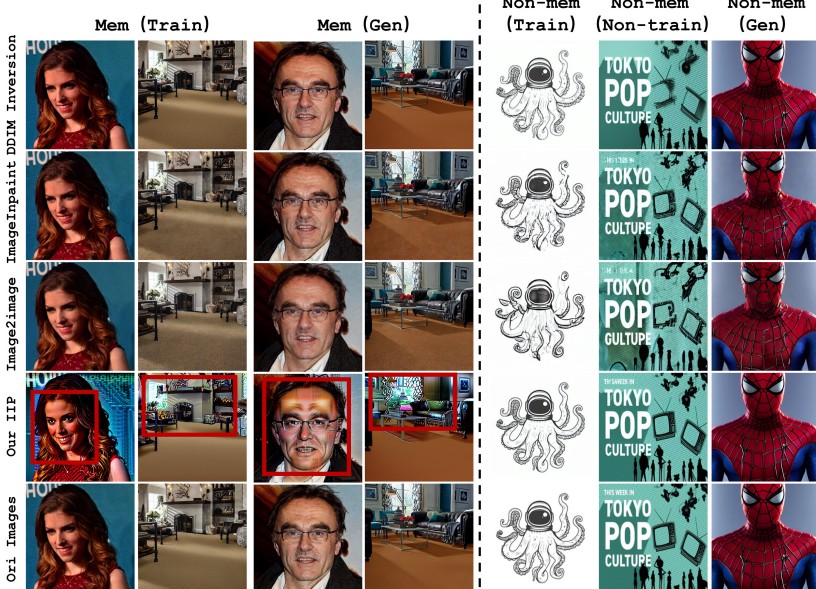

Figure 6: Qualitative comparisons with baseline methods under different settings. Our IIP demonstrates the most distinguishable results between memorized and non-memorized images (red boxes).

| Method | AUC | | | | Acc. | | | | TPR@1%FPR | | | |
|---|---|---|---|---|---|---|---|---|---|---|---|---|
| | #2K | #4K | #8K | AVG | #2K | #4K | #8K | AVG | #2K | #4K | #8K | AVG |
| DDIM Inversion | 0.739 | 0.741 | 0.733 | 0.738 | 0.681 | 0.677 | 0.674 | 0.677 | 0.062 | 0.062 | 0.075 | 0.066 |
| Inpaint | 0.919 | 0.921 | 0.919 | 0.920 | 0.879 | 0.875 | 0.871 | 0.875 | 0.621 | 0.640 | 0.623 | 0.628 |
| Image-to-image | 0.932 | 0.934 | 0.933 | 0.933 | 0.891 | 0.892 | 0.887 | 0.890 | 0.722 | 0.719 | 0.703 | 0.715 |
| SecMI | 0.631 | 0.630 | 0.629 | 0.630 | 0.614 | 0.610 | 0.608 | 0.611 | 0.061 | 0.049 | 0.057 | 0.056 |
| PIA | 0.710 | 0.711 | 0.710 | 0.710 | 0.653 | 0.649 | 0.651 | 0.651 | 0.074 | 0.098 | 0.096 | 0.089 |
| $\text{IIP}_{\text{sim}}$ | 0.964 | 0.965 | 0.963 | 0.964 | 0.935 | 0.939 | 0.936 | 0.937 | 0.866 | 0.867 | 0.866 | 0.866 |
| $\text{IIP}_{\text{mtcnp}}$ | **0.980** | **0.978** | **0.977** | **0.978** | **0.956** | **0.958** | **0.957** | **0.957** | **0.920** | **0.920** | **0.921** | **0.920** |

Table 3: Quantitative comparisons with baseline methods across varying sizes of generated datasets. Key results are highlighted according to Tab. 2.

as highlighted in red boxes. In comparison, non-memorized images closely resemble the originals. Overall, our method performs the best in distinguishing memorized images qualitatively.

### 4.2.2 QUANTITATIVE COMPARISON

To comprehensively evaluate the effectiveness of different methods, we conducted quantitative experiments across various datasets (Tab. 2 and 3). We utilized the $L_1$ distance to assess the similarity, which generally outperforms $L_2$ distance for most methods (see A for $L_2$ results). In addition to the similarity metric, our method can also employ the magnitude of TCNP as another metric.

We present the results for detecting memorized images from non-memorized training images and non-training images in Tab. 2. Among baseline methods DDIM Inversion is the most effective and achieves AUC values exceeding 0.9. However, its TPR@1%FPR values are low, similar to image inpainting and image-to-image, indicating in a relatively high miss rate for memorized images, SecMI and PIA, exhibit limited effectiveness, yielding the lowest AUC and TPR@1%FPR values. This may stem from that cues of memorization are less prominent for these methods. In contrast to other methods, our approach demonstrates nearly perfect discriminative performance, achieving AUC values exceeding 0.99, PR-AUC values greater than 0.96 for both metrics, and TPR@1%FPR values surpassing 0.98 for the similarity metric.

For detecting memorization on generated datasets (Tab. 3), we observe that image inpainting and image-to-image methods show notably improved performance, particularly in TPR@1% FPR, with the best results exceeding 0.7. SecMI and PIA also show slight improvements. This may be attributed to the generated images being potentially influenced by the model, which may activate intermediate outputs. However, DDIM Inversion diminishes its effectiveness, as both generated memorized and non-memorized images are more easily recoverable after inversion. Nonetheless, our method consistently demonstrates superior performance, with both AUC and Acc. exceeding 0.9 for

| Method | Mem (Train) / Non-mem (Train) | | | Mem (Train) / Non-mem (Non-train) | | | Mem (Gen) / Non-mem (Gen) | | |
|---|---|---|---|---|---|---|---|---|---|
| | AUC | AUC-PR | TPR@1%FPR | AUC | AUC-PR | TPR@1%FPR | AUC | Acc. | TPR@1%FPR |
| DDIM Inversion | 0.847 | 0.303 | 0.046 | 0.865 | 0.345 | 0.046 | 0.792 | 0.732 | 0.084 |
| Inpaint | 0.901 | 0.495 | 0.092 | 0.891 | 0.494 | 0.092 | 0.618 | 0.597 | 0.035 |
| Image-to-image | 0.562 | 0.114 | 0.000 | 0.510 | 0.103 | 0.000 | 0.642 | 0.609 | 0.043 |
| SecMI | 0.525 | 0.110 | 0.000 | 0.513 | 0.107 | 0.000 | 0.534 | 0.523 | 0.019 |
| PIA | 0.508 | 0.107 | 0.000 | 0.504 | 0.104 | 0.000 | 0.621 | 0.586 | 0.041 |
| $IIP_{sim}$ | **0.996** | **0.959** | **0.822** | **0.998** | **0.984** | **0.914** | 0.892 | 0.819 | 0.452 |
| $IIP_{mtcnp}$ | 0.823 | 0.667 | 0.480 | 0.838 | 0.699 | 0.539 | **0.918** | **0.844** | **0.636** |

Table 4: Quantitative comparisons with baseline methods across different datasets under attacks.

two metrics, and the results of MTCNP showing TPR@1% FPR surpassing 0.9. Interestingly, we observe that the similarity metric is more effective in detecting memorized training images, while the MTCNP performs better on the generated dataset. This discrepancy may stem from model consistency, as generated images tend to evoke memorization characteristics more readily and are more easily reconstructed.

### 4.2.3 PERFORMANCE OF DIFFERENT METHODS UNDER DATA AUGMENTATION ATTACKS

We conduct experiments on various methods under different data augmentation attacks, including random flip, rotation, and color changes, widely used data augmentation techniques for training or image preprocessing, to evaluate the robustness of different methods for detection. The results in Table 4 show that most methods exhibit a reduction in performance. For detecting memorization from non-memorized training and non-training datasets, only DDIM Inversion, image inpainting, and our approach retain effectiveness, with AUC values exceeding 0.6. However, these methods exhibit TPR@1%FPR values lower than 0.1, and their AUC-PR scores are also below 0.5, indicating a significant probability of false positives. In contrast, our method, based on the similarity metric, demonstrates the best overall performance and exhibits superior robustness against data augmentation attacks, Additionally, for detecting memorized images from generated images, DDIM Inversion shows good robustness in terms of AUC but with the TPR@1%FPR value falling below 0.1. Other methods all suffer significant performance declines. However, our approach maintains the best performance, with the MTCNP metric demonstrating enhanced robustness for generated images. The two metrics complement each other, resulting in our method exhibiting the most stable and robust performance against data augmentation attacks.

| Component | Method | Mem (Train) / Non-mem (Train) | | | Mem (Gen) / Non-mem (Gen) | | |
|---|---|---|---|---|---|---|---|
| | | AUC | AUC-PR | TPR@1%FPR | AUC | Acc. | TPR@1%FPR |
| w/o both | Sim. | 0.938 | 0.699 | 0.388 | 0.665 | 0.636 | 0.182 |
| w/o N.P.P. | Sim. | 0.993 | 0.936 | 0.796 | 0.738 | 0.681 | 0.058 |
| w/o S.L.C. | Sim. | 0.970 | 0.837 | 0.664 | 0.919 | 0.886 | 0.730 |
| | MTCNP | 0.871 | 0.428 | 0.184 | 0.944 | 0.879 | 0.513 |
| Full Model | Sim. | **0.999** | **0.995** | **0.987** | **0.964** | **0.935** | **0.866** |
| | MTCNP | **0.995** | **0.962** | **0.842** | **0.980** | **0.956** | **0.920** |

Table 5: Ablation studies of different components in our IIP. The N.P.P. denotes the non-memorized prompt perturbation module and the S.L.C. denotes semantic latent code.

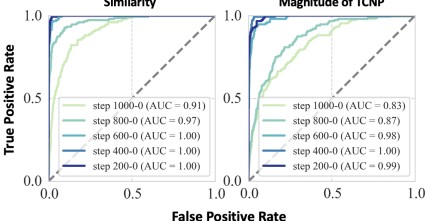

Figure 7: Effects of perturbations.

### 4.3 ABLATION STUDY

We conducted the ablation study on both the training and generated datasets, and the results are present in Tab. 5. We observe that with the introduction of semantic latent codes, the performance of our method improved across both datasets. Furthermore, the non-memorized prompt perturbation module not only facilitates the availability of the MTCNP metric but also improves the effectiveness of the similarity metric. However, due to insufficient semantic information to activate memorization clues, the TPR@1%FPR values for both the similarity and MTCNP metrics remain relatively low. Whereas, the full model exhibits the best performance.

We also analyze our performance with initializations at different timesteps on the training dataset in Fig. 7. The similarity metric demonstrates better performance in detecting memorized training images. Both the two metrics exhibit improvements from *step 1000* to *step 400*. Specifically, the similarity metric can nearly identify all memorized images starting after *step 600*, while the magnitude of TCNP shows a slight decrease in the AUC value starting at *step 200*. This may be attributed to minor distribution differences between the training and generated memorized images.

## 5 RELATED WORK

### 5.1 TEXT-TO-IMAGE DIFFUSION MODELS

Recently, pre-trained models have achieved great success in real-world applications (Lin et al., 2024b; Wu et al., 2024b; Lin et al., 2024a; Zhang et al., 2024; Xiong et al., 2024; Chen et al., 2025). Especially, with the development of large image-text paired datasets (Schuhmann et al., 2022; 2021) and diffusion models (Ho et al., 2020; Song et al., 2020; Sohl-Dickstein et al., 2015), text-conditioned generation based on diffusion models has sparked significant interest in text-conditioned generation for various image-generation tasks (Hertz et al., 2022; Zhang et al., 2023c; Brooks et al., 2023). Stable Diffusion, one of the most widely used diffusion models, demonstrates impressive fidelity and diversity. SD has been integrated into many downstream models like ControlNet (Zhang et al., 2023a) and DreamBooth (Ruiz et al., 2023), driving advancements in generative tasks.

### 5.2 MEMORIZATION IN DIFFUSION MODELS

Carlini et al. (2023); Somepalli et al. (2023a;b) have revealed the widely employed text-to-image diffusion model, Stable Diffusion (Rombach et al., 2022) and Imagen (Saharia et al., 2022) can reproduce training data during generation, which includes not only exact duplications but also general compositions from the training samples (Webster, 2023). Carlini et al. (2023) extract memorized image-text pairs by generating a great number of images with various prompts and searching for the most similar training images. Webster (2023) propose white and black attacks to extract memorized image-text pairs by the amount of noise modification in one-step synthesis and the edge consistency in generated images. Recently research has focused on detecting memorization from prompts. Wen et al. (2024) propose that the magnitude of predicted noise and effectively detect memorized prompts. The clue of MTCNP has been a significant source of inspiration for our work. Ren et al. (2024) find that the distribution of cross-attention exhibits great differences between memorized and non-memorized prompts. Ma et al. (2024) pioneered the analysis of detecting memorized images in diffusion models without original prompts and proposed an adaptive algorithm for this task. However, their work focuses solely on exact memorization with a limited number of memorized images. Differing from the works mentioned above, we uncover two characteristics for image-level memorization detection and formalize the task across various datasets and real-world scenarios with a comprehensive experimental setup, enhancing practical applicability.

Membership inference attack (Shokri et al., 2017) can effectively distinguish between training and non-training images but struggles to differentiate between memorized and non-memorized training images, as both are classified simply as training data. For diffusion models, Matsumoto et al. (2023); Hu & Pang (2023) focus on unconditional diffusion models that do not require prompt access. Carlini et al. (2023) propose a query-based method using diffusion losses, while Kong et al. (2023) employ DDIM Inversion as the ground-truth trajectory to infer memberships. Dubiński et al. (2024) introduce a dataset for membership inference attacks on Stable Diffusion and propose an attack targeting process modification in diffusion models. Matsumoto et al. (2023); Hu & Pang (2023) focus on unconditional diffusion models that do not require prompt access. Unlike these membership inference attack methods, our approach focuses on determining whether an image is memorized by the model instead of used for training, without accessing the original prompts.

## 6 CONCLUSION

In this paper, we first reveal two key characteristics of memorized images under inference perturbation during the original generation period: 1) lower similarity the original images, and 2) larger magnitudes of TCNP. Based on these observations, we propose a novel image-level memorization detection framework IIP. We are motivated to derive significant characteristics. Specifically, we obtain the semantic latent code of original images using unconditional DDIM Inversion and optimize the non-memorized prompt embedding for perturbation in the subsequent inference stage. After perturbation, the memorized images would exhibit lower similarity to the original images and a larger magnitude of TCNP, which contrasts with the non-memorized images. These distinct characteristics facilitate our effective detection. Extensive experiments demonstrate the state-of-the-art performance of our method across different datasets in identifying both training and generated memorized images, even under data augmentation attacks.

ACKNOWLEDGMENTS

This work is supported by the National Natural Science Foundation of China (NSFC) under Grants No. 62272460, Beijing Natural Science Foundation under Grant No. 4232037and the Special Fund for Key Program of Science and Technology of Jiangsu Province under Grant No. BG2024042.

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

# A APPENDIX

## A.1 ADDTIONAL EXPERIMENTAL RESULTS ON SDV1.4.

In this section, we provide additional experimental results on SDv1.4.

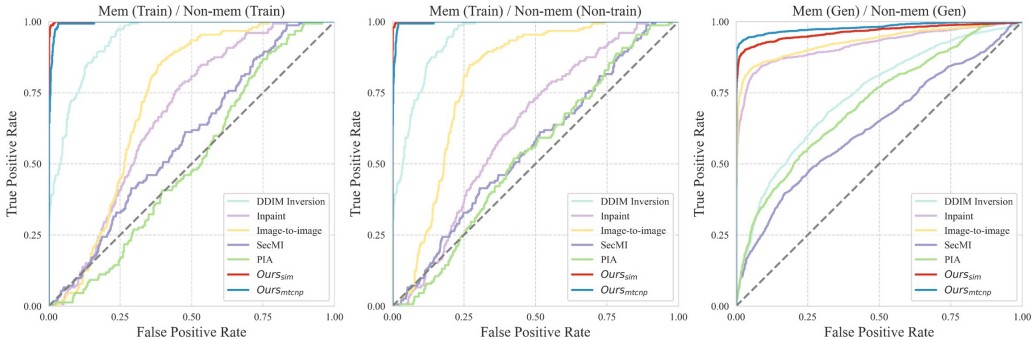

Figure A1: ROC curves of different methods for detecting memorized images from non-memorized training images, non-training images, and non-memorized generated images on SDv1.4.

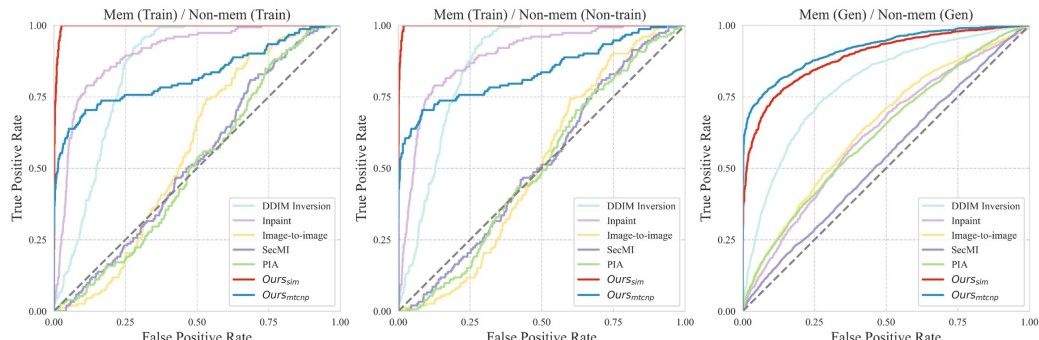

Figure A2: ROC curves of different methods on SDv1.4 under data augmentation attacks for detecting memorized images across different datasets as Fig. A1.

**Results of ROC Curves.** Fig. A1 and Fig. A2 present the ROC curves across different datasets and settings on SDv1.4. As shown in Fig. A1, we can observe that our method achieves near-perfect discrimination in detecting memorized training images from non-memorized training and non-training images, with the similarity metric demonstrating excellent effectiveness. For detecting memorization in generated images, the MTCNP metric exhibits relatively strong performance. Specifically, the similarity scores are more effective for identifying memorized training images, while MTCNP proves better for detecting memorization in generated images, which aligns with our findings in the paper. Furthermore, as shown in Fig. A2, our similarity-based metric demonstrates strong robustness against data augmentation attacks, even though the performance of MTCNP degrades considerably. While after data augmentation attacks on generated images, our method still achieves the best performance, with MTCNP maintaining superior effectiveness. The complementary strengths of these two metrics contribute to the overall robustness and effectiveness of our approach across various settings and data augmentation attacks.

**Results of $L_2$ similarity distance.** We further evaluate the effectiveness of various reconstruction-based methods using the $L_2$ distance, with results presented in Tab. A1 and Tab. A2. Our observations indicate that, except for image inpainting, all other methods exhibit a decline in performance when compared to $L_1$ scores. Notably, our method demonstrates a significant drop in TPR@1%FPR

values. This decline may be attributed to the more pronounced modifications to the image content introduced by the image inpainting method, whereas our approach primarily relies on subtle texture variations, rendering the $L_1$ distance a more effective metric. Despite the slight performance reduction observed under the $L_2$ distance, our method continues to outperform others and maintains state-of-the-art performance.

| Method | Mem (Train) / Non-mem (Train) | | | Mem (Train) / Non-mem (Non-train) | | | Mem (Gen) / Non-mem (Gen) | | |
|---|---|---|---|---|---|---|---|---|---|
| | AUC | AUC-PR | TPR@1%FPR | AUC | AUC-PR | TPR@1%FPR | AUC | Acc. | TPR@1%FPR |
| DDIM Inversion | 0.847 | 0.303 | 0.046 | 0.865 | 0.345 | 0.046 | 0.690 | 0.648 | 0.028 |
| Inpaint | 0.901 | 0.495 | 0.092 | 0.891 | 0.494 | 0.092 | 0.924 | 0.885 | 0.644 |
| Image-to-image | 0.562 | 0.114 | 0.000 | 0.510 | 0.103 | 0.000 | 0.928 | 0.890 | 0.715 |
| $\text{IIP}_{\text{sim}}$ | **0.996** | **0.959** | **0.822** | **0.998** | **0.984** | **0.914** | **0.960** | **0.919** | **0.815** |

Table A1: Quantitative results using $L_2$ distance to evaluate similarity for reconstruction-based methods on SDv1.4. The best and second-best results are highlighted in **bold** and underline.

| Method | Mem (Train) / Non-mem (Train) | | | Mem (Train) / Non-mem (Non-train) | | | Mem (Gen) / Non-mem (Gen) | | |
|---|---|---|---|---|---|---|---|---|---|
| | AUC | AUC-PR | TPR@1%FPR | AUC | AUC-PR | TPR@1%FPR | AUC | Acc. | TPR@1%FPR |
| DDIM Inversion | 0.842 | 0.298 | 0.053 | 0.861 | 0.346 | 0.059 | 0.788 | 0.723 | 0.088 |
| Inpaint | 0.718 | 0.176 | 0.007 | 0.685 | 0.160 | 0.013 | 0.630 | 0.603 | 0.042 |
| Image-to-image | 0.539 | 0.109 | 0.000 | 0.512 | 0.103 | 0.000 | 0.616 | 0.588 | 0.041 |
| $\text{IIP}_{\text{sim}}$ | **0.994** | **0.945** | **0.776** | **0.995** | **0.945** | **0.855** | **0.867** | **0.794** | **0.341** |

Table A2: Quantitative results using $L_2$ distance to evaluate similarity under data augmentation attacks for reconstruction-based methods on SDv1.4. Key results are highlighted according to Tab. A1.

**Addtional qualitative comparisons across different datasets.** We present additional qualitative results on SDv1.4 in Fig. A3, where significant similarity inconsistencies are highlighted within red boxes. The results indicate that our method exhibits a more pronounced distinction in similarity inconsistencies between memorized and non-memorized images.

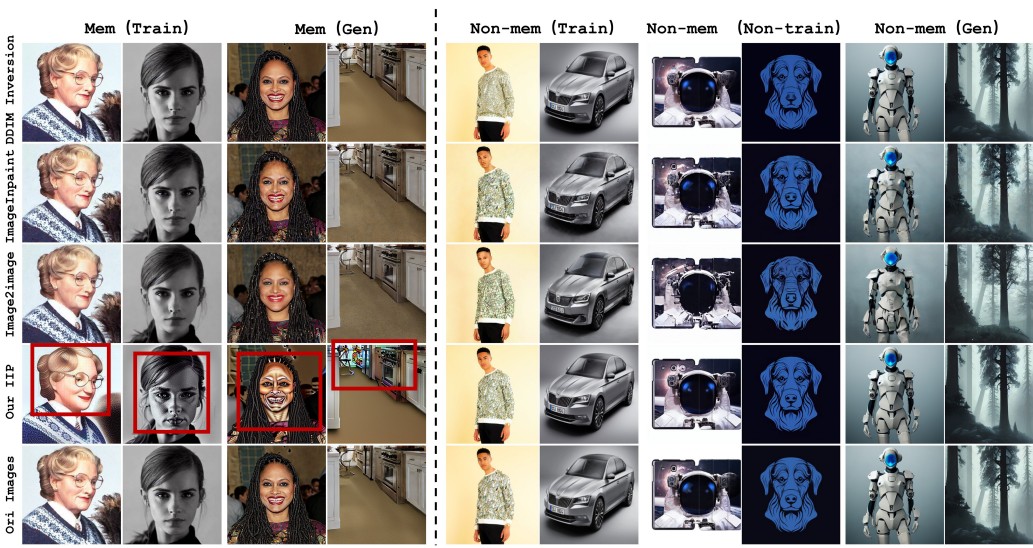

Figure A3: Addtional qualitative comparisons with baselines across different datasets on SDv1.4. Our method consistently achieves the best distinguishable results for memorization detection.

## A.2 QUANTITATIVE RESULTS ON OTHER VERSIONS OF STABLE DIFFUSION.

**Datasets.** To further assess the effectiveness of our method across different text-to-image diffusion models, particularly in detecting memorized training images, we conduct qualitative experiments on the training and non-training datasets for SDv1.5 and SDv2.1. For memorized training images, we follow Webster (2023) and collect 190 and 78 memorized images for SDv1.5 and SDv2.1, respectively. We obtain training images that have an SSCD similarity score (Pizzi et al., 2022) of generated results exceeding 0.7, strongly indicating duplication (Somepalli et al., 2023a;b). Furthermore, since the extracted training and non-member sets from LAION-mi (Dubiński et al., 2024) for SDv1.4 also apply to SDv1.5 and SDv2.1, we select 1200 non-memorized training images from the member sets of LAION-mi, with an SSCD similarity score below 0.15 of the memorized images, along with 1200 non-member images from LAION-mi as non-training images for these two Stable Diffusion models.

**Quantitative results and performance under data augmentation attacks.** We present the results for detecting memorized training images from non-memorized training and non-training images in Tab.A3 and Tab.A4, evaluated using our similarity metric. As shown, our method achieves AUC-PR values exceeding 0.96 and TPR@1%FPR above 0.87 across both models and settings, demonstrating our superior performance across different stable diffusion models. Furthermore, we analyze the performance under data augmentation attacks for SDv1.5, with results shown in Tab.A5. While baseline methods show significant accuracy degradation after attacks, our method maintains near-consistent performance, highlighting its exceptional robustness.

| Method | Mem(Train) / Non-mem(Train) | | | Mem(Train) / Non-mem(Non-train) | | |
|---|---|---|---|---|---|---|
| | AUC | AUC-PR | TPR@1%FPR | AUC | AUC-PR | TPR@1%FPR |
| DDIM Inversion | 0.876 | 0.689 | 0.474 | 0.882 | 0.697 | 0.505 |
| Inpaint | 0.572 | 0.145 | 0.005 | 0.515 | 0.131 | 0.000 |
| Image-to-image | 0.786 | 0.271 | 0.016 | 0.858 | 0.354 | 0.021 |
| SecMI | 0.508 | 0.141 | 0.011 | 0.546 | 0.160 | 0.016 |
| PIA | 0.594 | 0.176 | 0.021 | 0.639 | 0.212 | 0.042 |
| **IIP$_{sim}$** | **0.986** | **0.967** | **0.874** | **0.988** | **0.973** | **0.895** |
| **IIP$_{mtcnp}$** | 0.967 | 0.924 | 0.784 | 0.970 | 0.934 | 0.826 |

Table A3: Quantitative results for different methods of detecting training memorized images across different datasets on SDv1.5.

| Method | Mem(Train) / Non-mem(Train) | | | Mem(Train) / Non-mem(Non-train) | | |
|---|---|---|---|---|---|---|
| | AUC | AUC-PR | TPR@1%FPR | AUC | AUC-PR | TPR@1%FPR |
| DDIM Inversion | 0.883 | 0.307 | 0.064 | 0.892 | 0.320 | 0.128 |
| Inpaint | 0.651 | 0.107 | 0.000 | 0.612 | 0.099 | 0.000 |
| Image-to-image | 0.713 | 0.099 | 0.000 | 0.784 | 0.134 | 0.000 |
| SecMI | 0.552 | 0.078 | 0.000 | 0.533 | 0.074 | 0.000 |
| PIA | 0.538 | 0.117 | 0.077 | 0.564 | 0.126 | 0.077 |
| **IIP$_{sim}$** | **0.998** | **0.979** | **0.974** | **0.998** | **0.985** | **0.987** |
| **IIP$_{mtcnp}$** | 0.825 | 0.390 | 0.321 | 0.852 | 0.486 | 0.346 |

Table A4: Quantitative results for different methods of detecting training memorized images across different datasets on SDv2.1.

---

https://huggingface.co/stable-diffusion-v1-5/stable-diffusion-v1-5
https://huggingface.co/stabilityai/stable-diffusion-2-1-base

| Method | Mem(Train) / Non-mem(Train) | | | Mem(Train) / Non-mem(Non-train) | | |
|---|---|---|---|---|---|---|
| | AUC | AUC-PR | TPR@1%FPR | AUC | AUC-PR | TPR@1%FPR |
| DDIM Inversion | 0.745 | 0.282 | 0.032 | 0.756 | 0.300 | 0.032 |
| Inpaint | 0.811 | 0.410 | 0.053 | 0.793 | 0.410 | 0.053 |
| Image-to-image | 0.532 | 0.136 | 0.016 | 0.586 | 0.155 | 0.016 |
| SecMI | 0.520 | 0.133 | 0.000 | 0.534 | 0.138 | 0.000 |
| PIA | 0.567 | 0.152 | 0.000 | 0.582 | 0.161 | 0.005 |
| $IIP_{sim}$ | **0.983** | **0.957** | **0.853** | **0.986** | **0.969** | **0.884** |
| $IIP_{mtcnp}$ | 0.919 | 0.781 | 0.532 | 0.928 | 0.812 | 0.611 |

Table A5: Quantitative performance of different methods under data augmentation attacks across different datasets on SDv1.5.

## A.3 Experiments in Real-world Scenarios.

In practical situations, memorization varies in different degrees beyond exact duplication of the training images (Carlini et al., 2023; Somepalli et al., 2023a;b). For example, partial replication (Somepalli et al., 2023a), where generated images incorporate specific elements or fragments of existing images, is a common phenomenon in real-world scenarios and presents substantial privacy and copyright risks. To evaluate our performance in real-world scenarios, we expand the experiments to include a broader range of memorization situations. Specifically, we apply SSCD similarity score thresholds of 0.4 and 0.5, which indicate high pixel-level replication Somepalli et al. (2023b). These thresholds enabled the identification of a spectrum of memorization cases, from full duplication to partial replication. The numbers of memorized images extracted under different thresholds for different models are shown in Tab. A6.

| Number of Memorized Image | Model | Filter |
|---|---|---|
| 743 | SDv1.4 | SSCD > 0.5 |
| 1626 | SDv1.4 | SSCD > 0.4 |
| 601 | SDv2.1 | SSCD > 0.5 |
| 1676 | SDv2.1 | SSCD > 0.4 |

Table A6: The numbers of memorized images for different models and SSCD thresholds.

| Method | Mem(Train) / Non-mem(Train) | | | Mem(Train) / Non-mem(Non-train) | | |
|---|---|---|---|---|---|---|
| | AUC | AUC-PR | TPR@1%FPR | AUC | AUC-PR | TPR@1%FPR |
| DDIM Inversion | 0.839 | 0.776 | 0.292 | 0.836 | 0.776 | 0.312 |
| Inpaint | 0.604 | 0.418 | 0.008 | 0.552 | 0.391 | 0.005 |
| Image-to-image | 0.743 | 0.522 | 0.005 | 0.813 | 0.611 | 0.005 |
| SecMI | 0.510 | 0.415 | 0.019 | 0.516 | 0.421 | 0.016 |
| PIA | 0.543 | 0.424 | 0.020 | 0.582 | 0.473 | 0.034 |
| $IIP_{sim}$ | **0.984** | **0.978** | **0.774** | **0.988** | **0.984** | **0.812** |
| $IIP_{mtcnp}$ | 0.904 | 0.884 | 0.416 | 0.911 | 0.896 | 0.450 |

Table A7: Comparison of different methods on SDv1.4 with SSCD score(>0.5) filtered memorized images.

Experimental results are presented in Tables A8, A7, A9, and A10, corresponding to different SSCD thresholds across different stable diffusion models. In all experimental settings, the proposed IIP method consistently outperforms the baseline approaches, demonstrating superior performance. Specifically, both AUC and AUC-PR values exceed 0.9 across all configurations, highlighting the

| Method | Mem(Train) / Non-mem(Train) | | | Mem(Train) / Non-mem(Non-train) | | |
|---|---|---|---|---|---|---|
| | AUC | AUC-PR | TPR@1%FPR | AUC | AUC-PR | TPR@1%FPR |
| DDIM Inversion | 0.805 | 0.792 | 0.179 | 0.802 | 0.792 | 0.202 |
| Inpaint | 0.596 | 0.538 | 0.010 | 0.541 | 0.509 | 0.009 |
| Image-to-image | 0.740 | 0.633 | 0.003 | 0.809 | 0.710 | 0.003 |
| SecMI | 0.510 | 0.591 | 0.005 | 0.537 | 0.629 | 0.010 |
| PIA | 0.538 | 0.603 | 0.010 | 0.577 | 0.648 | 0.017 |
| $IIP_{sim}$ | **0.981** | **0.985** | **0.689** | **0.987** | **0.990** | **0.750** |
| $IIP_{mtcnp}$ | 0.850 | 0.895 | 0.260 | 0.860 | 0.905 | 0.287 |

Table A8: Comparison of different methods on SDv1.4 with SSCD score($>$0.4) filtered memorized images.

| Method | Mem(Train) / Non-mem(Train) | | | Mem(Train) / Non-mem(Non-train) | | |
|---|---|---|---|---|---|---|
| | AUC | AUC-PR | TPR@1%FPR | AUC | AUC-PR | TPR@1%FPR |
| DDIM Inversion | 0.768 | 0.599 | 0.077 | 0.779 | 0.611 | 0.101 |
| Inpaint | 0.599 | 0.402 | 0.012 | 0.555 | 0.377 | 0.012 |
| Image-to-image | 0.731 | 0.474 | 0.008 | 0.800 | 0.562 | 0.013 |
| SecMI | 0.582 | 0.440 | 0.045 | 0.559 | 0.415 | 0.020 |
| PIA | 0.505 | 0.384 | 0.042 | 0.530 | 0.414 | 0.045 |
| $IIP_{sim}$ | **0.955** | **0.935** | **0.624** | **0.956** | **0.943** | **0.737** |
| $IIP_{mtcnp}$ | 0.742 | 0.665 | 0.191 | 0.764 | 0.704 | 0.276 |

Table A9: Comparison of different methods on SDv2.1 with SSCD score($>$0.5) filtered memorized images.

| Method | Mem(Train) / Non-mem(Train) | | | Mem(Train) / Non-mem(Non-train) | | |
|---|---|---|---|---|---|---|
| | AUC | AUC-PR | TPR@1%FPR | AUC | AUC-PR | TPR@1%FPR |
| DDIM Inversion | 0.730 | 0.680 | 0.044 | 0.743 | 0.693 | 0.054 |
| Inpaint | 0.543 | 0.528 | 0.010 | 0.507 | 0.503 | 0.003 |
| Image-to-image | 0.747 | 0.657 | 0.009 | 0.813 | 0.732 | 0.015 |
| SecMI | 0.553 | 0.642 | 0.028 | 0.529 | 0.622 | 0.016 |
| PIA | 0.516 | 0.613 | 0.021 | 0.542 | 0.642 | 0.024 |
| $IIP_{sim}$ | **0.930** | **0.925** | **0.489** | **0.932** | **0.931** | **0.598** |
| $IIP_{mtcnp}$ | 0.704 | 0.781 | 0.116 | 0.730 | 0.808 | 0.177 |

Table A10: Comparison of different methods on SDv2.1 with SSCD score($>$0.4) filtered memorized images.

effectiveness of our method. These results further substantiate the performance of IIP in real-world scenarios where partial image replication may occur, making it particularly suitable for detecting broader memorization in text-to-image diffusion models.

## A.4    COMPARISON WITH DETECTING MEMORIZED PROMPTS (WEN ET AL., 2024).

We further clarify the difference between our inversion-based inference perturbation (IIP) and memorization mitigation techniques in (Wen et al., 2024), which propose MTCNP as an effective metric to detect memorized prompts. We make several new observations that extend this finding.

For memorization detection, Wen et al. (2024) finds memorized prompts derive larger MTCNP. We extend the observation that even when perturbed prompts (random or meaningless) are added during the inference process, the phenomenon of larger MTCNP still persists. This suggests that the MTCNP is not solely reliant on the original prompt, but also on some features related to the image itself, which we detailedly discussed Sec. 2.2.

Besides, while Wen et al. (2024) minimizes the magnitude of text-conditional predictions to perturb the original prompt embedding at the initial time step during generation. Our method first obtains the latent code by unconditional DDIM Inversion in latter timesteps and employs random and meaningless prompt for inference, the prompt embedding optimized cooperates with the latent code obtained from the inversion. We specify our optimization differences as follows:

- Different perturbing stages. We perturb the text embedding in the latter stage of inference (after timestep 30 in the experiments), while Wen et al. (2024) perturb the text embedding at the initial timestep.

- The minimization is processed with different textual prompts and latent codes. Our method uses random and meaningless prompts, cooperating with the latent code obtained from DDIM Inversion. In contrast, Wen et al. (2024) employs the original prompts and random gaussian latent code.

To further demonstrate the differences, we conducted experiments using the techniques in Wen et al. (2024): (a) perturbing text embedding in the initial timestep and (b) minimizing the magnitude of text-conditional predictions with original prompt and random gaussian latent code for the image-level memorization task.

| Method | AUC | AUC-PR | TPR@1%FPR |
|---|---|---|---|
| (a) perturb embedding (Wen et al., 2024) | 0.673 | 0.540 | 0.040 |
| (b) minimize the MTCNP (Wen et al., 2024) | 0.665 | 0.580 | 0.067 |
| $\textbf{IIP}_{sim}$ | 0.984 | 0.978 | 0.774 |
| $\textbf{IIP}_{mtcnp}$ | 0.904 | 0.884 | 0.416 |

Table A11: Qualitative results of techniques in (Wen et al., 2024) and IIP.

The experimental results demonstrate the different performance between our method and Wen et al. (2024). Our pipeline is proposed to effectively solve image-level memorization detection tasks.

Furthermore, the evaluation metric *similarity between the original and perturbed images* mainly depends on the inference perturbation pipeline and the optimization objective is alternative. MTCNP is one possible optimization objective in our current implementation, IIP can also employ objectives that do not rely on MTCNP. We employ the proposed similarity distance metric. Specifically, we replace the MTCNP-based loss function with similarity-based as follows:

$$L_{cd} = \frac{1}{t_I - t_{I'}} \lambda_d \cdot \sum_{t=t_{I'}}^{t_I} |z_t - \widetilde{z}_t| + \lambda_e \cdot ||E_t(c_d) - E_t(c_{null})||_2, \qquad (11)$$

where we exchang the MTCNP $||\epsilon_\theta(z'_t, t, c_d) - \epsilon_\theta(z'_t, t, c_{null})||_2$ in the original loss function 7 to similarity distance $|z_t - \widetilde{z}_t|$.

The experimental results are shown in Tab. A12. Results demonstrate that both optimization objectives perform comparably, indicating that the optimization objective is alternative and similarity computation is not heavily dependent on MTCNP.

## A.5 LIMITATIONS AND DISCUSSIONS.

**Limitations.** While our method demonstrates superior performance across various datasets, with different metrics performing optimally for different datasets, there are limitations under data augmentation attacks for generated datasets. Specifically, when faced with generated images with data

| Model | Memorized Images | IPP$_{sim}$ Results of Different Optimization Objectives | Mem(Train) / Non-mem(Train) | | | Mem(Train) / Non-mem(Non-train) | | |
|---|---|---|---|---|---|---|---|---|
| | | | AUC | AUC-PR | TPR@1%FPR | AUC | AUC-PR | TPR@1%FPR |
| SDv1.4 | SSCD > 0.5 | MTCNP | 0.984 | 0.978 | 0.774 | 0.988 | 0.984 | 0.812 |
| | | Sim. Distance | 0.981 | 0.975 | 0.777 | 0.983 | 0.977 | 0.802 |
| | SSCD > 0.4 | MTCNP | 0.981 | 0.985 | 0.689 | 0.987 | 0.990 | 0.750 |
| | | Sim. Distance | 0.975 | 0.982 | 0.676 | 0.978 | 0.984 | 0.696 |
| SDv2.1 | SSCD > 0.5 | MTCNP | 0.955 | 0.935 | 0.624 | 0.956 | 0.943 | 0.737 |
| | | Sim. Distance | 0.956 | 0.940 | 0.700 | 0.959 | 0.946 | 0.730 |
| | SSCD > 0.4 | MTCNP | 0.930 | 0.925 | 0.489 | 0.932 | 0.931 | 0.598 |
| | | Sim. Distance | 0.930 | 0.955 | 0.570 | 0.936 | 0.960 | 0.598 |

Table A12: Quantitative performance of different optimization objectives for IIP$_{sim}$.

augmentation attacks, our method may not perform as effectively. The robustness in detecting memorization from these defense-enhanced generated images remains an area in need of further improvement. Additionally, although our work primarily focuses on image-level memorization detection, dememorization during generation represents an important aspect that could further mitigate the retention of user content in the model's memory. This issue has not been addressed in the current study, but we intend to investigate it in future research.

**Ethics and Broader Impacts.** In recent years, the rise of generative models has increasingly highlighted issues related to copyright and data security. Memorization, as a form of strong training data duplication, poses significant threats to both model owners and users. Our method offers a straightforward and effective solution for users and regulators to verify whether images have been memorized by the model, in the absence of original prompts. Notably, our method demonstrates nearperfect detection capabilities for memorized training images and exhibits high robustness against data augmentation attacks, providing an effective tool for identifying data leaks and copyright threats from the perspective of memorization.

