# OpenReview forum: "Image-level Memorization Detection via Inversion-based Inference Perturbation"
_ICLR.cc/2025/Conference — ICLR 2025 Poster_

### Official Review · Reviewer_8dpK · 2024-10-31

**Soundness:** 2
**Presentation:** 2
**Contribution:** 3
**Rating:** 6
**Confidence:** 3

**Summary:**

The paper introduces a method for image-level memorization detection. It is motivated by an analysis showing two key characteristics of memorized images under perturbed inference: similarity discrepancy and a large magnitude of text-conditional noise prediction. The key idea of the method seems to be using an unconditional DDIM inversion to derive latent codes and optimizing non-memorized prompt embeddings for effective perturbation. The method is evaluated on a number of datasets, showing strong ability to detect memorized images.

**Strengths:**

* The results are strong and the method seems reliable in detecting memorized images
* There is an analysis section to motivate the approach and explain why it may work
* The paper is generally well-written and easy to read

**Weaknesses:**

* The prompts appear to be an input to the optimization procedure that generates the optimized prompt embedding, so even if the prompts are not directly paired with the images, we may not be able to say the method does not rely on prompts (as may seem from some parts of the paper). This can be discussed more to explain to what extent we do not need access to the original prompts.
* All settings are based on Stable Diffusion v1.4: while this is a common model for image generation, one would like to see if the same method works across different models. There are currently various image generation models available, so it would be good to try the method on more - e.g. two more - to see if this method generalizes more broadly.

Minor:
* Ori prompt would be easier to understand if it was written as original prompt or orig. prompt

**Questions:**

* To what extent are the perturbed prompts related to the original prompts? Can we really say we do not use the original prompts in any way? E.g. if we use something derived from them
* How expensive is it to run the method for an image to test if it is memorized?

---

### Official Review · Reviewer_dGYC · 2024-11-03

**Soundness:** 2
**Presentation:** 3
**Contribution:** 2
**Rating:** 6
**Confidence:** 3

**Summary:**

The authors propose an algorithm named IIP to detect memorized images generated by LDMs. The algorithm features two observations during DDIM generation with perturbed prompts (1) a notable similarity discrepancy (2) a large magnitude of text-conditional noise prediction.

**Strengths:**

(1) The observed phenomenon seems to be universal and statistically notable.

(2) Within the experiment setup of this paper, the algorithm shows promising performance.

(3) The presented figures are illustrative and depictive.

**Weaknesses:**

**Major**

(1) The biggest concern I have about this paper is its high overlap with existing work [1], making its technical and empirical contribution weak. I shall support my claim with the following facts: (a) The metric TCNP/MTCNP is proposed by [1], which also applies the metric to memorization detection. (b) Perturbing the textual embedding during DDIM generation is already used by [1] to mitigate memorization. (c) minimizing the magnitude of text-conditional prediction is already used by [1] to achieve the perturbed prompt embedding. It wasn't until I carefully re-read [1] that I noticed so much overlap between the proposed algorithm and an existing work. I believe the authors fail to give enough credit to the existing work, as all this necessary information is not shown in the submitted manuscript.

(2) The authors tend to use big words like "SOTA" and "pioneer". However, I am not fully convinced by the importance of the "image-level detection" setting proposed by the authors. If we can detect memorization as early as in the generation phase like [1], why would we bother doing the "image-level detection"?

[1] IMAGE-LEVEL MEMORIZATION DETECTION VIA INVERSION-BASED INFERENCE PERTURBATION; https://openreview.net/pdf?id=84n3UwkH7b

**Minor**

(1) Only SD v1.4 is considered, which is not enough because the performance of the proposed algorithm might be highly affected by the architecture, the training data, and the sampling configuration of the LDMs.

(2) Missing period in line 181 "during the subsequent generation process [period here] Interestingly..."

**Questions:**

Please see the weakness part.

---

### Official Review · Reviewer_JFcJ · 2024-11-03

**Soundness:** 3
**Presentation:** 3
**Contribution:** 2
**Rating:** 6
**Confidence:** 2

**Summary:**

The paper presents a new method for identifying whether an image was part of the diffusion model training set, without access to the prompt used for generation. As far as the authors know, they are the first to identify this task. To this end - inference under prompt perturbations is examined for memorized / not memorized samples, with the aid of DDIM's inversion capabilities (specifically unconditional inversion to avoid dependency on the prompt). Under this process, they find two key properties that differentiate between memorized and non-memorized samples. Building on their insights, they propose a Inversion-based Inference Perturbation (IIP) -  a novel method for the task at hand and out-perform the competitors on an extensive test suite.

**Strengths:**

1. The results present SOTA performance with respect to the defined task and the evaluated existing methods.

2. The experiments are extensive: Evaluate both success, sensitivity, and soundness.

3. The writing is good and articulate.

4. Novelty of the method: The authors propose for the first time to experiment with inference with perturbed prompts for the task at hand.

**Weaknesses:**

1. Novelty of the task: Please have a look at [1], [2] - these work engage in membership inference without using prompts, as far as I know. Could the authors clarify how their proposed task of image-level memorization detection differs from membership inference, particularly in light of the cited works that perform membership inference without prompts?

[1] Matsumoto, T., Miura, T., & Yanai, N. (2023, May). Membership inference attacks against diffusion models. In 2023 IEEE Security and Privacy Workshops (SPW) (pp. 77-83). IEEE.
‏

[2] Hu, H., & Pang, J. (2023, November). Loss and Likelihood Based Membership Inference of Diffusion Models. In International Conference on Information Security (pp. 121-141). Cham: Springer Nature Switzerland.

2. Comparison to other methods: Could the authors add comparisons to relevant methods that do not require prompt access (e.g. the methods mentioned above)?

3. Additional experiments with a different stable diffusion model would substantiate the proposed method's advantage.

4. Requiring inversion of the model is quite restrictive, considering how quickly generative technology changes, and how rare it is for generative models to enable such inversions. Can the authors think of ways adapt their findings to non-invertible scenarios?

5. Minor Weaknesses:
a. In all related figures - change "ori" to the full word "original".
b. The expression " images exhibit greater similarity discrepancy " in the introduction is rather confusing

**Questions:**

Please refer to Weaknesses above.

---

### Official Review · Reviewer_8Z99 · 2024-11-04

**Soundness:** 2
**Presentation:** 3
**Contribution:** 2
**Rating:** 6
**Confidence:** 4

**Summary:**

This paper investigates the memorization limitation of the Stable Diffusion model to help protect the training data’s privacy. The paper proposes a new task: image-level memorization detection, which is different from existing works that detect memorization during inference. Then, based on two insights that memorized images under perturbed inference have a notable similarity discrepancy and a large magnitude of text-conditional noise prediction, the paper proposes IIP framework that uses unconditional DDIM inversion to derive latent noises for the images and conducting perturbations. The paper also construct a setup for this new task and demonstrated better performance than baselines.

**Strengths:**

1. This work tackles an important and practical problem: diffusion models can memorize their training data, potentially leading to privacy concerns. These issues are thoroughly discussed and effectively motivated in the paper.
2. The paper is well-presented, clearly structured, and easy to follow.
3. This paper further proposes a new task of image-level memorization detection and a correspondingly designed method for this new task.

**Weaknesses:**

1. The motivation for the proposed new image-level memorization detection task is based on a misunderstanding of the related work. Specifically, in lines 46-47, the paper claims that “However, these approaches rely heavily on access to the original prompts, which is often impractical.” Since all baselines and this paper experiment on the text-to-image Stable Diffusion model, using text prompts is practical. Also, the baseline methods do not need an organized prompt list that contains triggering prompts since they can obtain detection signals (such as TCNP in [1]) during the inference process. They can then detect the potential memorized images being generated within only one inference step and with good accuracy. Such lines of method are actually more practical than the proposed image-level memorization detection task, as the memorization is detected and halted from the source (way before the image is even generated). Thus, the proposed task is of limited practical significance. I would suggest the authors consider investigating unconditional diffusion models (where there is no text prompt) and see if the proposed method works.
2. The finding that large TCNP correlates with memorization is not novel. I understand that the paper differs in performing DDIM inversion to the clean image, however, it is actually the identical finding of [1] that a noise can present large TCNP even after the first step of DDIM denoising.

[1] Detecting, Explaining, and Mitigating Memorization in Diffusion Models.

**Questions:**

Please see the weaknesses regarding motivation and novelty.

---

### Public Comment · ~Zhe_Ma3 · 2024-11-28

The proposed image-level memorization detection task and the inversion-based detection idea, which are the main motivation of this work and considered novel by the reviewers, are also proposed in [1]. However, I didn't see adequate discussion on the connection or difference between the two works, but only a short and point-missing description of [1] in Sec 5.2.

[1] Zhe Ma, et al. Could It Be Generated? Towards Practical Analysis of Memorization in Text-To-Image Diffusion Models. arXiv preprint arXiv:2405.05846, 2024.

---

> ### Author Response · Authors · 2024-11-28
> **To Zhe Ma**
>
> > Thanks for your attention and comments on our work. We would like to clarify the main difference with [1] as follows:
>
> > **Memorization Scope and Comprehensive Setup**
> > - Our work addresses **both exact duplication and partial replication**, expanding the scope of memorization detection. The reviewers agree that exploring **various degrees of image-level memorization** detection substantially enhances the practical importance of our work.
> > - We have made significant efforts to establish a comprehensive evaluation setup for image-level memorization detection. Specifically:
> >   - We have constructed a much **larger and more practical** benchmark, simulating diverse real-world memorization scenarios. This includes memorized images under varying degrees of replication, ensuring a more practically meaningful evaluation.
> >   - Additionally, we extend our approach to detect memorized images from both training and non-training images, as well as robustness evaluation under **data augmentation attacks**, to better address real-world scenarios.
> > - In contrast, [1] focuses solely on exact duplication and conducts its experiments with only 78 memorized images and 100 non-memorized images. As denoted by Reviewer dGYC and Reviewer 8Z99, this setting has **limited practical significance** because only a relatively small subset of training images is likely to be memorized.
> > - In summary, our primary contribution lies in establishing this comprehensive setup for image-level memorization detection, which we believe will foster broader discussions on infringement inspection in text-to-image diffusion models, addressing real-world scenarios and practical applications.
>
> > **Different Motivation**
> > - While both methods involve inversion-based detection, the approach in [1] and our proposed Inversion-based Inference Perturbation (IIP) differ significantly in their objectives and implementation.
> >   - In [1], the focus is on using inversion to **derive model prediction errors** as clues for memorization. Specifically, [1] optimizes the distribution of noise $\epsilon$ and further compares it to a Gaussian distribution to evaluate whether memorization has occurred.
> >   - In contrast, our motivation is to leverage inversion to obtain **latent codes at smaller timesteps** that retain rich semantic information about the original images. This enables us to detect memorization by analyzing distinct patterns in both the similarity of the original and reconstructed images, as well as the magnitude of text-conditional noise predictions (MTCNP) after perturbing the inference stage.
>
> > **New Detection Clues and Novel IIP Framework**
> > - We discover that memorized images exhibit **lower similarity to their original versions** and maintain **larger magnitudes of text-conditional noise predictions (MTCNP)** after perturbing the generation procedure.
> > - Building on these findings, we propose our **Inversion-based Inference Perturbation (IIP)** framework, which first employs DDIM inversion to derive **latent codes** containing rich semantic information from the original images and then **optimizes random prompt embeddings** to perturb the inference process effectively.
> > - In contrast, [1] focuses on the clue **model's prediction error** and **compares it on the target images to safe unconditional models.** —none of which are part of the motivation or core of our method. Our approach is developed independently of [1] and introduces a novel perspective on memorization detection.
>
>
> > **Effectiveness and Efficiency**
> > - As mentioned earlier, we have constructed comprehensive setups to rigorously demonstrate the **effectiveness** of our IIP framework. Specifically, we evaluate it on SD v1.4, v1.5, and v2.1, using **hundreds to thousands of memorized images** across **various degrees** of memorization. This extensive validation ensures the effectiveness and applicability of our IIP method in real-world scenarios.
> > - Additionally, our IIP framework is **highly efficient**, processing a single image in just **3.5 seconds**. However, the prompt inversion in [1] seems considerably more time-consuming.
> > - Moreover, Our method does not rely on other models while [1] involves distribution comparisons with *unconditional model*.
>
>
> > Acknowledgment:
> > - We would like to express our gratitude for this comment. We will revise the discussion of [1] in the next version to provide a clearer comparison.
> > - In addition, we look forward to the open-source release of [1], which will enable further comparisons and improvements in future work.

---

> ### Public Comment · ~Zhe_Ma3 · 2024-11-29
>
> It has nothing to do with the specific methods or evaluation settings, which won't be taken into consideration until a research problem is defined. The concentration should be the motivation of this work, i.e., how do we reach such a problem. The motivation of this work seems exactly the same as Ma et al.
>
> What I'm really concerned with is that, although the authors have noticed previous work conducted by Ma et al., who clearly pointed out the limitation of prompt-based analysis methods and proposed the image-only setting, only a short description is included in line 512, Sec 5.2: "Ma et al. (2024) further analyze memorization across conditional and unconditional diffusion models". This description is vague and misses the main point of their work. Also, it should not be stated together with the prompt-based methods.
>
> I request that the authors clarify the problem-level relationship of these two works.

---

> ### Author Response · Authors · 2024-11-29
> **To Zhe Ma**
>
> > Thank you for your continued feedback. While we acknowledge the similarity in the research problem, we would like to clarify that **specific methods and evaluation settings play a significant role** in assessing the contributions of a research work. We emphasize that our work **addresses a broader range of memorization** including both exact duplication and partial replication. This significantly extends the **applicability and practical importance** of our task and detection method. Additionally, the two works are distinct in detection clues and the framework, as well as effectiveness and efficiency under extensive experimental evaluations, as outlined in our previous response.
>
> > However, we recognize the value of [1].  As the paper’s revision deadline has passed, we will ensure to include a clearer explanation in the next version, **highlighting its main point and fully discussing it**.
>
> > Thank you again for directly pointing out your concerns and for your interest in our work—this truly reflects the value of open peer review.

---

### Public Comment · ~Gunjan_Dhanuka1 · 2025-03-19
**Missing Code Repository for the paper**

Dear Authors,

Congratulations on your paper's acceptance to ICLR 2025! I am very interested in reproducing your results and exploring the ideas further. However, I noticed that the GitHub repository mentioned in the paper is not working anymore, and I haven’t received a response to my email which I sent to the corresponding authors about the same.

Could you kindly share the code or provide an updated link to the repository? It would greatly help in advancing research in this direction.

Thank you!

---

> ### Public Comment · ~Yue_Jiang3 · 2025-03-19
> **Thanks for your interest**
>
> Thank you for your interest in our work! The GitHub repository is correct—we have just opened this repository. We are also in the process of organizing the dataset and code and will update them as soon as possible. Thanks!

---

> > ### Public Comment · ~Gunjan_Dhanuka1 · 2025-03-24
> > **To Authors**
> >
> > Thanks for opening the repository, I am looking forward to the code and datasets.

---

### Meta-Review · Area_Chair_NKbp · 2024-12-16

**Metareview:**

This paper presents a new setting of image-level memorization detection given a user-provided reference image. It is generally well-received and has promising experimental results.

The authors are encouraged to improve the paper in the final version regarding the following aspects.
- clearer distinctions from related work
- broader experimental validation across different models (experiments from the rebuttal should be included)
- a more compelling discussion of the practical significance of the task
- a (nice to have) discussion about how possibly removing the detected image-level memorization

Although some margins need to be improved, this paper still stands with its merits. So it is recommended that it be accepted.

**Additional Comments On Reviewer Discussion:**

It is apparent that the rebuttal discussions were adequate and addressed most concerns of the reviewers, resulting in a consistent final rating.

---

### Decision · Program_Chairs · 2025-01-22

Accept (Poster)